# Epstein–Barr Virus: How Its Lytic Phase Contributes to Oncogenesis

**DOI:** 10.3390/microorganisms8111824

**Published:** 2020-11-19

**Authors:** Quincy Rosemarie, Bill Sugden

**Affiliations:** Department of Oncology, McArdle Laboratory for Cancer Research, School of Medicine and Public Health, University of Wisconsin-Madison, Madison, WI 53705, USA; rosemarie@wisc.edu

**Keywords:** EBV, lytic reactivation, lytic phase, oncogenesis, tumor survival, immune evasion

## Abstract

Epstein–Barr Virus (EBV) contributes to the development of lymphoid and epithelial malignancies. While EBV’s latent phase is more commonly associated with EBV-associated malignancies, there is increasing evidence that EBV’s lytic phase plays a role in EBV-mediated oncogenesis. The lytic phase contributes to oncogenesis primarily in two ways: (1) the production of infectious particles to infect more cells, and (2) the regulation of cellular oncogenic pathways, both cell autonomously and non-cell autonomously. The production of infectious particles requires the completion of the lytic phase. However, the regulation of cellular oncogenic pathways can be mediated by an incomplete (abortive) lytic phase, in which early lytic gene products contribute substantially, whereas late lytic products are largely dispensable. In this review, we discuss the evidence of EBV’s lytic phase contributing to oncogenesis and the role it plays in tumor formation and progression, as well as summarize known mechanisms by which EBV lytic products regulate oncogenic pathways. Understanding the contribution of EBV’s lytic phase to oncogenesis will help design ways to target it to treat EBV-associated malignancies.

## 1. Introduction

EBV-positive malignancies have been associated with the latent phase of EBV’s life cycle; a non-productive phase in which no progeny virus is formed. However, there is now increasing evidence that EBV’s lytic phase contributes to EBV oncogenesis (reviewed in [1,2,3,4,5]).

What is meant by the term “EBV’s lytic phase”? To orient the reader, here we introduce the terminology that has developed with the study of EBV. Epstein–Barr Virus (EBV) is a human γ-herpesvirus that infects a variety of cells in vivo and naïve B-lymphocytes efficiently in vitro. EBV induces and maintains proliferation of the infected B cells. In these cells, EBV remains latent, with few viral genes being expressed reminiscent of the lysogenic state of some bacteriophage (reviewed in [6]). On rare occasions, these infected cells alter their transcription to support EBV’s productive cycle, in which progeny virus particles are produced. Early studies of clones of infected cells found on the order of 1 per 10^3^ cells to 1 per 10^6^ cells would support the productive cycle per 24 h [7]. Even with its rarity, EBV’s productive cycle has been well examined, being essential for the propagation of this human tumor virus. These studies have led to a distinctive terminology. The complete productive cycle of EBV is termed “lytic” because productively infected cells eventually lyse. The lytic phase has been divided into “immediate early”, “early”, and “late” phase by analogy to the detailed analyses of α-herpesviruses [8,9]. Viral genes that support viral DNA amplification are classified as early and those that support viral particle formation are termed late. The early lytic genes have been further divided, with the subset that mediates the transcription of the viral DNA replication genes being termed “immediate early”. Careful studies of viral gene expression during infection of primary B cells have led to the realization that multiple immediate early and early lytic genes of EBV are expressed during the first few days following infection, while others are introduced as proteins from the tegument of the viral particles. These include the immediate early genes *BZLF1* and *BRLF1*, transcriptional activators that regulate viral and cellular transcription [10,11,12]. The expression of these early lytic genes following infection of primary B cells does not lead to a complete lytic cycle [11] and reactivation of the lytic phase where one or more early lytic genes are expressed in the absence of virus production is now termed the “abortive lytic phase” [13].

In this review, we will discuss the evidence of EBV’s lytic phase contributing to tumorigenesis, and the mechanisms by which it does so.

## 2. EBV’s Lytic Phase Contributes to Tumorigenesis by Production of Infectious Viral Particles

No EBV-positive tumor forms without a precursor cell being infected by this virus. What evidence indicates that the risk of EBV’s oncogenesis is dependent on the amount of infectious EBV in a person? In other words, are people who support atypically high levels of EBV’s productive cycle or who are infected by variants of EBV that support entry into the lytic phase efficiently more likely to develop EBV-positive tumors? Multiple findings are consistent with viral load being proportional to the risk of EBV-associated malignancy. First, a large, prospective serological survey was conducted in Uganda between 1972 and 1979 to test for an association between infection with EBV and the development of Burkitt Lymphoma (BL) [14,15]. Blood samples were taken from 42,000 children, and the children were monitored for development of BL. Over the course of the survey, 16 children developed BL (1 lymphoma/2600 children). These children were found to have significantly elevated titers of the antibodies to EBV’s viral capsid antigen (VCA) prior to the manifestation of their BLs [15]. Thus, infection with EBV was a risk factor for developing Burkitt Lymphoma in these children and their immune response to components of the virus particle prior to lymphoma formation correlated with this risk factor. One evident interpretation of this finding is that a higher viral load increases the risk of children developing Burkitt Lymphoma.

A second finding which indicates that an increasing viral load increases the risk of developing Burkitt Lymphoma comes from an appreciation of malaria being a co-factor for the risk of its development. Denis Burkitt first hypothesized that malaria might be a risk factor for this cancer, based on the geographical distribution of BL overlapping with the presence of holoendemic malaria in central Africa [16]. Insightful work on the role of T cells in suppressing the growth of EBV-infected cells revealed that during the acute phase of the malarial infectious cycle, the number of anti-EBV cytotoxic T cells is reduced significantly [17]. This insight was further illuminated by a study in The Gambia where Burkitt Lymphoma is endemic. Lam et al. [18] measured the number of EBV-infected B cells in the blood of children during the acute phase of malaria and found five-fold more infected cells than in the blood of convalescent children. They also showed that when naïve B cells were co-cultivated with EBV-positive B cells from acute phase children, there was spontaneous outgrowth of lymphoblastoid cells. However, this outgrowth was absent when the cells were treated with acyclovir, which blocks lytic DNA replication and the subsequent production of progeny viruses [18]. These combined findings indicate that there is an increased number of EBV-transformed B cells in vivo during the acute phases of malaria, and this increase arises at least in part from the release of EBV from infected cells leading to new infections of uninfected B cells.

The increased numbers of EBV-infected cells constitute an increased risk for developing Burkitt Lymphoma because they provide a larger reservoir in which mutations that also drive oncogenesis can occur. For example, greater than 90% of Burkitt Lymphoma tumor cells have a translocation between the *c-Myc* locus and one of the three immunoglobulin loci [19,20,21]. The translocation places the expression of the *c-Myc* proto-oncogene under the control of an immunoglobulin locus, resulting in its constitutive expression in B cells which express immunoglobulin genes [22]. Thus, increased viral loads in vivo lead to increased numbers of EBV-infected cells, which, as proliferating lymphoblasts, can both undergo and maintain chromosomal translocations that can contribute to EBV’s oncogenesis. The increased viral loads in vivo would thus contribute to an increased risk for developing Burkitt Lymphoma.

A third kind of evidence to support an increased viral load adding to the risk for developing an EBV-caused cancer comes from studies of naturally occurring or engineered variants of EBV that differ in their support of the lytic phase. This evidence is complex, though. With the advent of assays for the activity of EBV in cell culture, researchers have screened for cells that release more particles of infectious EBV [23] and new strains of EBV that could support entry into the lytic phase more efficiently than others [24,25,26,27]. Multiple studies have subsequently identified sequence variations in the promoter of *BZLF1*, an immediate early lytic gene which regulates entry into the lytic phase. These variants can affect entry of EBV into its lytic phase and the efficiency with which they transform B cells in vitro. For example, one triple mutant within the *BZLF1* promoter increases the rate of virus release from transfected 293 cells by 7–9-fold and reduces the efficiency of viral transformation of primary B cells by 10-fold [28]. One naturally occurring variant within the *BZLF1* promoter, termed Zp-V3, which can be bound by the NFAT transcription factor, increases an infected B cell’s responsiveness to entry into the lytic phase by treatment with anti-Ig antibodies, but does not affect the efficiency of transformation of primary B cells [29]. This variant is found in approximately 50% of Burkitt Lymphomas (ibid.) and has no apparent effect on tumor formation in immunocompromised mice following reconstitution with human cord blood cells [30].

Typically, entry and completion of the lytic phase results in cell death, thus seemingly incompatible with tumorigenesis. However, half of surveyed Burkitt Lymphomas have the lytic responsive Zp-V3 variant, and clearly these infected cells can evolve to become lymphomas. Therefore, we conclude that following infection of a cell, more efficient entry into EBV’s complete lytic phase is not a barrier to oncogenesis. What might explain this seeming discrepancy? Entry and passage of newly infected B cells through the complete lytic phase is undetectable for the first 12 days following infection [31]. This absence is a consequence of the incoming viral DNA being unmethylated [32] and that BZLF1 preferentially binds to methylated sites [33,34]. Over time, EBV viral DNA in infected cells becomes increasingly methylated, eventually supporting BZLF1 binding on specific sites which drive the complete lytic phase. Given that during the first few days following infection the infected cells proliferate unimpeded, when a progeny cell eventually supports the complete lytic phase and dies, its siblings can continue to proliferate, thus supporting tumorigenesis.

It does seem likely that variants of EBV that support their lytic phase more efficiently than do others would be more likely to yield higher viral loads in people and accordingly have an increased risk of being oncogenic. This likelihood is supported by the observations that isolates with the Zp-V3 variant of the *BZLF1* promoter occur twice as frequently in Burkitt Lymphomas and nasopharyngeal carcinomas as in non-malignant EBV-infected cells from people in the same regions of the world in which these cancers are found [29].

## 3. EBV’s Lytic Gene Expression in EBV-Associated Tumor Samples

That high viral load is a risk factor for developing an EBV malignancy and that variants of EBV that support entry into the lytic phase may contribute to this risk are indirect findings for understanding how EBV’s lytic phase might affect tumor cells. Examination of EBV’s gene expression in samples of EBV-associated tumors has identified the expression of EBV lytic genes in tumor cells. The most common lytic gene assayed, the immediate early gene *BZLF1*, has been identified in Burkitt Lymphoma (BL) [35], nasopharyngeal carcinoma (NPC) [36,37], and gastric carcinoma (GC) [38] samples. Most of these identifications used RT-PCR to detect either unspliced or spliced variants of *BZLF1*. In several studies, subsets of the tumor samples were also subjected to BZLF1 immunohistochemistry (IHC) [35,36]. The IHCs confirmed the RT-PCR results for each tumor, with BZLF1-expressing cells comprising 10–50% of tumor cells in each BZLF1-positive sample. Given that IHCs specifically detect proteins, these results indicate that, in those samples tested, not only are lytic genes transcribed, the corresponding proteins are indeed synthesized.

BZLF1 is a principal activator of the cascade of EBV’s lytic gene expression, but its detection does not ensure completion of the viral lytic cascade as shown by its detection in newly infected B cells, which do not progress to lytic reactivation [10,11,12]. Indeed, assays for additional lytic genes have often failed to detect several such key genes in tumor samples expressing BZLF1. In one study of eight NPC biopsies, RT-PCR assays for the expression of early genes *BMLF1* and *BBLF2/3* (encoding EBV’s M transactivator and primase-associated factor, respectively) found that only five of eight samples tested positive for *BMLF1* expression, and none tested positive for *BBLF2/3* [39]. All eight NPC samples had previously tested positive for *BZLF1* expression [36], thus revealing that EBV-associated tumors often harbor cells undergoing an incomplete or abortive lytic phase, in which some early lytic genes are expressed, but no viral particles are produced [13].

Surprisingly, despite evidence for an incomplete lytic phase, several studies have shown that late lytic genes are expressed in tumor samples [38,39]. The expression of late lytic genes requires EBV genome amplification in *cis* [40], and DNA amplification itself requires a minimum set of early lytic genes encoding the core replication machinery: *BALF5*, *BALF2*, *BBLF2/3*, *BBLF4*, *BMLF1*, *BSLF1*, and *BMRF1* [41,42]. However, some of the key components of this core replication machinery have not been detected in tumors in which the late lytic genes are expressed. These findings constitute a conundrum: how can late lytic genes be expressed in the absence of expression of viral genes shown to be required for their expression? Some late lytic genes are “leaky”, having low-level expression early in the lytic phase and are further upregulated following lytic DNA replication [43]. It is possible that these leaky late genes constitute some of the late genes detected in tumor samples lacking DNA replication. Indeed, *BALF4* (encoding glycoprotein B) was consistently detected in GC samples lacking in expression of some DNA replication genes [38], and is a leaky late gene [43]. However, leaky late genes do not underlie the expression of *BLLF1* [39], *BNRF1*, and *BPLF1* [38], which are true late genes and therefore dependent upon lytic DNA replication [43]. The answer to this puzzle may lie in the inefficient detection of the genes required for lytic DNA replication or in their being expressed in only a small subset of the tumor cells. What is clear is that EBV-associated tumors often express an assortment of lytic genes spanning the early and late lytic gene sets. It also appears that the frequent expression of some lytic genes in EBV-associated tumor samples supports their potential contributions to EBV’s oncogenesis. In the next section, we discuss what these contributions might be.

## 4. Functional Contributions of EBV’s Lytic Genes to EBV’s Oncogenesis

Several studies have used animal models to uncover the importance of EBV’s lytic phase to lymphomagenesis. One study by Hong et al. [44], used engineered variants, mutated for entry into the lytic phase, to investigate their tumorigenesis and growth in mice with severe combined immunodeficiency (SCID). The authors used mutant variants of EBV with knockouts of the early lytic genes, *BZLF1* (BZLF1-KO) and *BRLF1* (BRLF1-KO) and infected human peripheral blood mononuclear cells (PBMCs) to generate BZLF1-KO and BRLF1-KO lymphoblastoid cell lines (LCLs). When grown in vitro, these mutant LCLs had similar proliferation rates as LCLs generated using wild-type EBV. However, when these LCLs were injected subcutaneously into the flanks of SCID mice, both BZLF1-KO and BRLF1-KO LCLs had reduced tumor growth rates compared to wild-type LCLs. Within groups of donor-matched PBMCs, the mutant LCLs grew to at most 25% of the volume of their wild-type counterparts, with most injections resulting in minimal or no tumor growth. The authors also conducted a rescue experiment using BZLF1-KO LCLs stably transfected with *BZLF1* expression vectors, in which *BZLF1* expression rescued the growth defect of BZLF1-KO LCLs upon injection into SCID mice. Thus, it is apparent that EBV’s early lytic phase does contribute to lymphomagenesis in SCID mice.

A second study demonstrated the role of lytic phase in lymphomagenesis in an infection model of humanized mice. Ma et al. [45] used a humanized NOD/LtSz-scid/IL2Rγnull (hNSG) mouse model, in which NSG mice were injected with purified human CD34+ cells to reconstitute the human immune system. These hNSG mice were infected with BZLF1-KO EBV intraperitoneally and assessed for lymphomas at 60 to 65 days post infection. The tumor incidence of mice infected with BZLF1-KO EBV was 14% (2/14), while that of wild-type EBV was 54% (6/11). Thus, consistent with the findings in SCID mice [44], variants of EBV defective for entry into the lytic phase are impaired for tumor growth in mice. In comparison to SCID mice, hNSG mice have increased immune responses, including functional T cells and B cells. The results from this study therefore indicate that contributions of EBV’s lytic phase to lymphomagenesis can occur in the presence of increased immune responses mediated by the reconstitution of the human immune system.

A third study that supports a role for EBV’s lytic phase in tumorigenesis analyzed primary effusion lymphomas (PEL) in humanized NSG mice [46]. PEL has been primarily associated with KSHV, but 90% of PELs are dually infected with EBV. McHugh et al. developed a dual-infection model in hNSG mice using wild-type or BZLF1-KO EBV, in conjunction with wild-type KSHV. They observed that mice infected with BZLF1-KO EBV + KSHV had a significantly decreased tumor incidence and multiplicity compared to mice infected with wild-type EBV + KSHV. This finding again highlights the significance of EBV’s entry into the lytic phase for its lymphomagenesis. The authors also noted that there is no difference in KSHV persistence in tumor cells dually infected with the wild-type vs. BZLF1-KO variants of EBV; this observation indicates that the role of EBV’s lytic phase likely involves more than a requirement for KSHV maintenance within these PEL-like cells.

Several additional insights can be gained from these mouse models to shed light on the mechanisms by which EBV’s lytic phase contributes to lymphomagenesis. One role of the lytic phase is to produce and release new infectious particles. To test whether this function is essential for lymphomagenesis, Hong et al. [44] did an additional experiment in which they injected SCID mice with wild-type LCLs and treated them with acyclovir (ACV), an inhibitor of viral lytic DNA replication. The doses of ACV treatment were determined such that the mean plasma concentration of ACV in the mice was comparable to the ACV doses used to inhibit the complete lytic phase in patients with infectious mononucleosis. The authors concluded that ACV treatment did not impair growth of the wild-type LCLs in SCID mice. This conclusion indicates that the complete lytic phase is not required for lymphomagenesis. It is also supported by work showing that EBV’s plasma viral load in infected mice was largely undetectable regardless of BZLF1-KO or wild-type virus variant status [45].

A separate study in NOD/Shi-scid-ILR2γ^null^ (NOD) mice further supports this conclusion [47]. In this study, the authors generated LCLs null for EBV’s *BALF5* gene encoding its DNA polymerase and therefore incapable of supporting lytic DNA amplification. When these LCLs, termed dBALF5, were xenografted into NOD mice, the mice were observed to have increased weight loss and lower survival rates in comparison to mice having xenografts infected with wild-type EBV. dBALF5 mice were also found to have higher EBV copies in various organs. In cells infected with wild-type EBV, an increased level of EBV copy number could be mediated by increased lytic DNA replication and/or the proliferation and accumulation of latently infected cells [48]. As EBV is unable to undergo lytic replication in the absence of *BALF5*, the increased EBV copies in dBALF5 mice is evidently caused by the proliferation of latently infected B cells. Furthermore, IHC examinations of dBALF5 organs revealed an increase in BZLF1-expressing cells, indicating an increased frequency of EBV-infected cells entering the lytic phase. Of note, dBALF5 EBVs are clearly capable of lytic reactivation; however, as they are unable to amplify their genome, we can infer that they are accordingly incapable of expressing late genes, as true late gene expression requires lytic amplification of the genome in *cis* [40]. Combined with findings from the BZLF1-KO studies and given that a complete lytic phase would lead to host cell lysis, this study confirms that EBV’s early lytic phase is important in tumorigenesis, whereas the late lytic phase is dispensable and even likely to inhibit tumor progression. This conclusion is also supported by EBV genome analysis of T/NK cells in chronic active EBV infection (CAEBV). Among the analyzed CAEBV patient samples, 35% (27/77) harbored intragenic deletions, with 77% (21/27) of these deleted genomes carrying deletions in regions essential for the production of infectious viral particles [47]. These deleted regions include the BART miRNA clusters (which contain miRNAs that negatively regulate *BZLF1* and *BRLF1*), the core lytic replication genes, as well as other genes essential for infectious particle production. Similar findings were found in cases of EBV-positive diffuse large B cell lymphomas (DLBCLs) and extranodal NK/T cell lymphomas (ENKLs) [47]. These analyses do highlight the potential roles of BZLF1, BRLF1, and early EBV genes in viral oncogenesis.

## 5. Cellular Regulation of Tumorigenesis by EBV’s Lytic Phase

EBV’s lytic phase affects tumorigenesis by modulating cellular pathways that influence tumor cells and tumor microenvironment. The following sections discuss multiple mechanisms by which EBV’s lytic phase influences tumorigenesis through the regulation of immunomodulation and immune evasion, angiogenesis and invasion, genomic instability, as well as cell cycle and apoptosis. For a summary of the EBV lytic genes included in this review and the tumorigenic properties they regulate, see Table 1.

### 5.1. Immunomodulation and Immune Evasion

EBV inhibits the immune response during its lytic phase to support the maturation and release of progeny virions. How might these activities contribute to tumor cells evading detection and elimination by the immune system? One mechanism of EBV’s immunomodulation during its lytic phase is by the induction of cellular cytokines such as IL-6, IL-8, IL-10, IL-13, and IL-1β [44,49,50,51,54,70]. These cytokines are known to support tumorigenesis, and IL-6, and IL-13 in particular have been shown to support growth and survival of EBV-associated tumors [51,70]. The increased expression of these cytokines has been shown to be mediated by several lytic proteins, including BZLF1, BRLF1, and BLLF3 (for details on the specific cytokines upregulated by these lytic proteins, see Table 1). The mechanism by which BZLF1 upregulates cytokine expression likely involves its function as transcriptional activator. BZLF1 binds to promoters of cytokine genes at AP-1 and/or ZRE sites, activating transcription of downstream gene [49,50,51]. BZLF1 is capable of binding nucleosomal DNA, increasing local chromosome accessibility through its interactions with chromatin remodeling enzymes, consistent with its designation as a pioneer transcription factor [71]. This induction of local open chromatin allows for a permissive state for transcription, through which BZLF1 could activate expression of its target genes, including cytokines. BRLF1’s mechanism of upregulating cytokine has not been directly examined, though we can speculate that its upregulation of cytokine gene expression is mediated by its transcriptional activator function. *BLLF3* is an early lytic gene encoding the viral dUTPase; how it upregulates cytokine expression is unclear.

In addition to regulating cellular cytokine expressions, EBV has its own IL-10 homolog: vIL-10, encoded by *BCRF1*. *BCRF1* is late lytic gene, although it is also expressed immediately after infection of primary B cells [55]. In the context of primary infection, vIL-10 has been shown to inhibit natural killer (NK) cell-mediated elimination of EBV-infected B cells and inhibit CD4+ T cell activities (ibid.). These functions support tumor cell evasion of immune detection, one of the hallmarks of cancer. Were EBV-infected cells to express BCRF1 transiently and survive, this expression could contribute to tumorigenesis.

EBV’s lytic phase has also been shown to accommodate immune evasion through prevention of antigen presentation. Several lytic gene products downregulate MHCs, including BGLF5, BILF1, BNLF2a, BZLF2, and BDLF3. BGLF5 is EBV’s alkaline exonuclease which mediates host shut off via mRNA degradation, leading to the inhibition of MHC synthesis [52]. BILF1 is a G protein-coupled receptor (GPCR) which inhibits MHC trafficking, thereby reducing immune recognition of host cells [53]. BNLF2a is an inhibitor of the transporter for antigen processing (TAP), which reduces recognition of infected cells by CD-8 T cells [55]. BZLF2, encoding the viral glycoprotein 42 (gp42) inhibits MHC II-mediated antigen presentation to T cells [57]. BDLF3, encoding glycoprotein 150 (gp150) ubiquitinates and downregulates MHCI and MHCII [56]. *BGLF5*, *BILF1*, and *BNLF2a* are early lytic genes, and we speculate that cells surviving the abortive lytic phase avoid immune detection by expressing these viral genes. On the other hand, *BZLF2* and *BDLF3* are late lytic genes, and their potential contributions to tumorigenesis would require their being expressed during the progression of the tumor cells. Overall, it seems that several lytic proteins could contribute to immune modulation and evasion by inducing inflammatory cytokines, inhibiting immune cell responses against EBV-infected primary B cells, and/or downregulating antigen presentation through MHCs.

### 5.2. Angiogenesis and Invasion

EBV is known to induce angiogenesis and extracellular matrix (ECM) degradation, leading to enhanced tumorigenesis and tumor invasion in NPCs. The role of EBV in angiogenesis is well documented in its latency phase (reviewed in [72]). In contrast, the angiogenic properties of EBV’s lytic phase are less well investigated. Nevertheless, early passage LCLs lacking BZLF1 (BZLF1-KO) or BRLF1 (BRLF1-KO) have been shown to have lower VEGF secretion compared to wild type [73]. VEGF (vascular endothelial growth factor) is a known stimulant of blood vessel formation. Correspondingly, supernatants from BZLF1-KO LCLs, when incubated with human dermal microvascular endothelial cells (HDMECs), induced less vessel formation in comparison to supernatants from wild-type LCLs, confirming that LCLs defective for lytic phase entry have decreased angiogenesis. This decrease is likely mediated by reduced VEGF production, although whether BZLF1 or BRLF1 directly regulate expression of VEGF is currently unclear.

EBV in its lytic phase also regulates the matrix metalloproteinases (MMPs). MMPs are involved in both angiogenesis and the degradation of ECM (reviewed in [74]). Angiogenesis and ECM breakdown, along with epithelial–mesenchymal transition (EMT), are processes that underlie tumor invasion and metastasis. BZLF1, and to some extent BRLF1, have been shown to be important for the upregulation of several MMPs in NPCs. BZLF1 directly upregulates MMP1, MMP3, and MMP9 by binding to AP-1 sites on the promoters of these genes [58,59,60]. All three MMPs contribute to cell invasiveness, as demonstrated in Matrigel invasion assays [59,60], and MMP3 has also been shown to promote cell migration [60]. Similarly, BRLF1 has been shown to upregulate MMP9, leading to pro-invasive effects in NPC cell lines [61].

Curiously, in B cells, BZLF1 has been shown to regulate the production of TIMP-1 (tissue inhibitor of metalloproteinases 1), a known broad-spectrum inhibitor of MMPs, during early EBV infection of primary B cells [75]. In early infection of EBV, BZLF1 expression increases in the days following infection, as does TIMP-1 in both its expression and secretion. Ectopic expression of BZLF1 in EBV-negative BL showed a corresponding upregulation of TIMP-1, whereas shRNA against BZLF1 in LCLs led to a reduction in TIMP-1 production. BZLF1 was found to bind AP-1 sites on the TIMP-1 promoter, indicating that it directly upregulates TIMP-1 expression and secretion (ibid.). Whether BZLF1 also regulates TIMP-1 in the context of lytic phase remains to be investigated. TIMP-1 downregulates MMP activities, and therefore is typically known to be anti-angiogenic (reviewed in [76]). However, it also has been shown to have an anti-apoptotic function, as demonstrated in LCLs treated with cisplatin [75], therefore providing tumor cells a survival advantage. Although both TIMP-1 and MMPs are induced by BZLF1, their induction has been measured individually in B cells or epithelial cells, respectively. Their seemingly contradictory effects may therefore reflect functions of BZLF1 differing in different host cells. In addition, it is currently unclear how TIMP-1 and MMPs interact in the context of the lytic phase, and thus the effects of this interplay on tumorigenesis remain unclear.

It is apparent that EBV’s lytic phase contributes to the modulation of angiogenesis and cell invasion. This contribution is mainly carried out through BZLF1 and BRLF1 as examined in cell culture, although there may also be other lytic factors involved. Generally, only a small subset of cells in an EBV-associated tumor undergoes the lytic phase. It is therefore apparent that this minor cell population, if it does influence tumor progression in vitro, does so non-cell autonomously by changing the tumor microenvironment and supporting angiogenesis and cell invasion.

### 5.3. Genomic Instability

EBV’s lytic phase has been shown to induce cellular genomic instability, one of the enabling characteristics of oncogenesis [77]. In NPC cell lines, chemical induction of the lytic phase in EBV-positive cells led to an increase in cells with micronuclei and γH2AX, markers of DNA damage and genomic instability [78]. Additionally, inhibition of the lytic phase through siRNA knockdown of BZLF1 reduced the number of γH2AX-positive cells upon chemical treatment, confirming the role of the lytic phase in the induction of DNA damage (ibid.). Similarly, in B cells, the lytic phase has also been associated with genomic instability. Infection of primary B cells with wild-type EBV led to an increased number of centrioles, formation of micronuclei, multinucleated cells, and/or aneuploidy, while infection with a lytic-defective variant (ΔZR; BZLF1 and BRLF1 double knockout) led to fewer cells with these genomic aberrations [66]. Accordingly, B cells infected with either variant of the virus had similar growth rates in vitro, but the ΔZR-infected LCLs had reduced tumor incidence and growth rate when injected into NSG mice. Surprisingly, genome-less virus-like particles (VLPs) were found to induce centriole amplification and aneuploidy upon contact with B cells and epithelial cells, regardless of their EBV status. Apparently, EBV does not need to establish stable infection to induce genomic aberrations. Moreover, this phenomenon was abolished in VLPs lacking the gp110 protein encoded by *BALF4*, a late lytic gene, implicating its contribution to the VLP-mediated genomic effects.

Several EBV lytic proteins have been shown to promote genomic instability. BNRF1, EBV’s major tegument protein, has been shown to be an important factor in inducing chromosomal instability. For example, Shumilov et al. found that infection with *BNRF1*-deleted (ΔBNRF1) EBV led to a decrease in the number of cells with genetic aberrations as compared to that of wild-type EBV [66]. The mechanism of BNRF1-mediated chromosomal instability is unclear, although sucrose gradient fractionation experiments found BNRF1 to be enriched in the centrosomal fraction. Another lytic protein, BGLF5, encoding the viral DNase, induced genomic instability upon transfection into EBV-negative epithelial cells [65]. Its mechanism of action seems to involve induction of DNA damage, correlating with its nuclease activity, and inhibition of DNA damage repair through decreased expression levels of DNA-repair genes including *MSH2*, *MSH6*, *MLH1*, and *PMS2* (ibid.). BALF3, EBV’s terminase protein, when transfected into EBV-negative NPC cell lines, led to increased γH2AX signal, formation of micronuclei, and accumulation of chromosomal aberration [62]. This induction of DNA damage and genomic instability was found to be dependent upon its endonuclease function. BGLF4, encoding the viral S/T kinase, has been shown to induce premature chromosome condensation [63,64]. This phenotype was found to be dependent on BGLF4′s kinase function through interactions with condensin and topoisomerase II [63]. BGLF4 was also found to induce the formation of micronuclei, a common marker of genomic instability [64]. Genomic instability predisposes cells to acquiring hallmarks of cancer. However, it is also important that these cells continue to proliferate as tumor cells. BGLF4 phosphorylates many cellular proteins including the Aurora kinases, ATM, CDK1, and CDC20, resulting in the activation of the DNA damage response pathways and inhibition of anaphase promoting complex/cyclosome (APC/C), both events typically associated with mitotic inhibition [79]. However, while high levels of BGLF4 in mammalian cell lines were shown to inhibit cell cycle progression, low levels were found to allow for completion of mitosis [64]. Therefore, while BGLF4 can induce genomic instability, its role in cell cycle and tumorigenesis is currently unclear and may be dependent on its expression level.

The role of EBV’s lytic phase in inducing genomic instability poses a potential inconsistency. Typically, cells that undergo the lytic phase die following the release of viral particles. This death means that any genomic aberrations induced in these cells become irrelevant to tumor progression. How could the lytic phase contribute to tumorigenesis by inducing genomic instability? One possible mechanism is through the VLP-mediated genomic instability uncovered by Shumilov et al. [66]. While the lytic cells die, the virions they produce could interact with neighboring cells and mediate a non-cell-autonomous effect, resulting in increased genomic instability in the tumor population. This hypothesis could also explain how a small percentage of lytic cells would have wide-reaching effects within a population of tumor cells. Another potential mechanism involves an incomplete lytic phase, also referred to as an abortive lytic phase. Except for BALF4, all the proteins implicated in inducing genomic instability are early lytic proteins. Thus, in the event of an abortive lytic phase, these early proteins could still be expressed and exert their function. An abortive lytic phase could increase the importance of early EBV proteins in viral tumorigenesis because it allows these cells to continue to proliferate and contribute to the tumor population. To test the role of an abortive lytic phase in tumorigenesis, it would be necessary to measure its frequency and function in vitro. Without such measurements, the extent to which an abortive lytic phase affects EBV’s tumorigenesis is currently unclear.

### 5.4. Cell Cycle Regulation and Apoptosis

EBV encodes two viral homologs of the cellular BCL2 anti-apoptotic proteins: BALF1 and BHRF1. These *vBcl-2* genes are classified as early lytic genes. However, their expression have also been detected early during infection, although at much lower levels than those found in the lytic phase, and have been found to be important for cellular transformation of B cells [67,68]. The expression of *BALF1* and *BHRF1* early during infection is independent of both BZLF1 and EBNA2, as virus variants lacking either gene supported expression of these anti-apoptotic genes as does wild-type EBV (ibid.). Additionally, BHRF1 has been shown to be expressed constitutively in Wp-restricted BLs, and at low levels in established LCLs two to four months post infection [68,69]. This long-term expression of BHRF1 is unlikely to be due to low level lytic reactivation in the cell population, as LCLs established using BZLF1-KO EBV were also found to express BHRF1 long term [68].

Both BALF1 and BHRF1 have been reported to have pro-survival, anti-apoptotic functions, although BALF1 has also been reported to have pro-death functions instead [80]. In an infection setting, EBV lacking both BALF1 and BHRF1 failed to generate LCLs and accumulated high levels of subG1 and apoptotic cells [67]. This defect is rescued in the presence of either one of these vBcl-2 genes, indicating their functional redundancy in supporting the transformation of B cells. Expression of *BHRF1* in both EBV-positive (Akata-BL) and EBV-negative (BL41, Eµ Myc mouse lymphoma) cell lines protected cells from death induced by treatment with ionomycin or etoposide [67]. This anti-apoptotic function of BHRF1 has been ascribed to its binding with and inhibition of the cellular pro-apoptotic protein, BIM [81]. However, BHRF1 was found to inhibit cell death even in the absence of BIM, indicating that its protective function is not mediated solely by its interaction with BIM [69]. In addition to BIM, BHRF1 also binds the pro-apoptotic PUMA and BAK proteins [69,82], showing that it may function through inhibition of several cellular BCL-2 family members.

In addition to its importance in the initial transformation of B cells, it appears that BHRF1 also contributes to the tumorigenicity and long-term survival of tumor cells for two reasons. First, expression of BHRF1 in Eµ-Myc hematopoietic stem and progenitor cells (HSPCs), followed by their injection into lethally-irradiated mice, led to an accelerated development of lymphomas [69]. Second, while BL cell lines typically have high rates of TP53 mutations, BL lines that naturally express BHRF1, including Sal-BL, Oku-BL, and Ava-BL, have wild-type TP53. This finding is consistent with BHRF1′s anti-apoptotic function compensating for the selective pressure to mutate TP53 in lymphomas (ibid.).

While *BALF1* and *BHRF1* are highly expressed in the lytic phase, it is evident that they also contribute to tumorigenesis early in infection and during latency. Whether by transient expression post-infection or Wp-mediated long-term expression, it is clear that the anti-apoptotic functions of these vBcl-2 genes are essential in ensuring cell survival and proliferation, two key characteristics of tumor cells.

In summary, EBV’s lytic phase regulates cellular oncogenic pathways, including those promoting angiogenesis, immunomodulation and immune evasion, genomic instability, as well as cell cycle and survival. These regulated pathways contribute to tumor formation and progression via cell-autonomous and non-cell-autonomous functions. Cell-autonomous functions that contribute to tumorigenesis occur in cells that undergo an incomplete (abortive) lytic phase; those cells that complete the lytic phase would die and can no longer contribute to the tumor. These functions include increased proliferation, immune evasion, and genomic instability, along with decreased apoptosis. On the other hand, non-cell-autonomous events can influence surrounding cells and foster a pro-tumorigenic microenvironment through angiogenesis, modifications of the extracellular matrix, and cytokine productions. Collectively, these cell-autonomous and non-cell-autonomous outcomes contribute to EBV-mediated oncogenesis.

## 6. EBV’s Lytic miRNAs in Tumorigenesis

In addition to its lytic proteins, EBV also regulates tumorigenesis through its miRNAs. EBV encodes two clusters of miRNAs, one in the BHRF1 locus and one in the BART locus. The BART miRNAs are detected at all phases of EBV’s life cycle, while the BHRF1 miRNAs are not detected in some cells in culture and are when the same cells are induced to enter their lytic phase [83]. The levels of one of the BHRF1 miRNAs also correlates with viral load in the blood of patients consistent with it being expressed preferentially during EBV’s lytic phase [84]. The BHRF1 miRNAs are found immediately after infection of primary B cells and in some cases of post-transplant lymphoproliferative disease [85,86]. The BART miRNAs contribute to transformation by regulating expression of multiple cellular genes as well as inhibiting immune recognition of the infected cell [87,88,89]. Their continued presence in cells in EBV’s lytic phase is likely to contribute also to the success of this portion of the viral life cycle. The new expression of the BHRF1 miRNAs in cells induced into EBV’s early lytic phase should affect this phase distinctively, though. Can any of these BHRF1 miRNAs also contribute to EBV’s oncogenesis?

miR-BHRF1-2 inhibits the tumor suppressors, PTEN and PRDM1, which would likely foster EBV’s tumorigenesis [86,90]. This same miRNA also inhibits expression of the IL-1 receptor 1 to limit signaling via receptor engagement and, potentially, any resulting inflammatory response [91]. Some of the targets of the BHRF1 miRNAs clearly affect late stages of EBV’s lytic phase, too. For example, miR-BHRF1-1 targets the ubiquitin ligase, RNF4, leading to an accumulation of SUMOylated viral proteins and completion of the lytic phase [92]. It is possible that an inhibition of RNF4 and of other cellular targets that aid the late stages of the lytic phase could also favor EBV’s oncogenesis; whether they do so now is uncertain.

EBV’s miRNAs do contribute to oncogenesis but these contributions are not easily classified as reflecting their expression during the latent or lytic phases of the viral life cycle. Most viral miRNAs are expressed during both phases. The BHRF1 miRNAs are expressed most efficiently during the lytic phase, though, and clearly also can foster EBV’s oncogenesis.

## 7. Inhibitor Studies: A Test for a Role for EBV’s Lytic Phase in Oncogenesis?

One essential role for EBV’s lytic phase in EBV’s oncogenesis is the production of infectious virus, which supports the infection and transformation of cells that subsequently can evolve into tumors. This role was inadvertently highlighted by the transplantation of tissues from allogeneic donors to recipients who would reject the grafts immunologically. Transplant physicians worked to limit the graft rejection by inhibiting the recipient’s immune response with, for example, cyclosporine A and anti-thymocyte globulin. Cyclosporine A acts by binding Cyclophilin A, which together inhibit the phosphatase activity of calcineurin. This phosphatase is needed for the activation of the NFAT transcription factor and much T-cell signaling [93]. Anti-thymocyte globulin kills a variety of T cells. One unanticipated consequence of this immunological inhibition was a dramatic increase in EBV-associated lymphoproliferative disease variously termed LPD or PTLD for post-transplant lymphoproliferative disease. PTLD could be rapidly fatal. In a study of 257 patients who received allogenic stem-cell transplantation, half died over the course of 17 years and 14% of these had developed PTLD unequivocally associated with EBV infection [94]. Their clinical symptoms did not foreshadow their disease adequately to allow treatment (ibid.). More recent work has found that limiting the level of immune suppression has greatly reduced the frequency of PTLD in hematopoietic cell transplant recipients while still fostering the reactivation of EBV as measured by the detection of its DNA in the serum of these transplant recipients [95].

What facets of EBV’s lytic phase contribute to oncogenesis after infection and transformation? A possible experimental route to address this question is to test small-molecule inhibitors of distinct steps within EBV’s lytic phase.

One prominent small-molecule inhibitor, acyclovir, was developed for treatment of Herpes Simplex Viruses (HSV) and is extremely effective. It acts by being preferentially phosphorylated by the HSV thymidine kinase, the product of which preferentially inhibits HSV DNA polymerases [96]. Acyclovir and its derivatives have been developed to inhibit other herpesviruses including EBV and some do block EBV’s DNA synthesis during its lytic phase. They, however, have not been found to be effective clinically [97,98]. This failure reflects the rarity of EBV-positive tumor cells supporting the complete lytic phase. Investigators have tried to surmount this difficulty by treating these tumors to induce the lytic phase and then using small-molecule inhibitors of viral DNA synthesis to kill the cells. This approach has been tested both in cell culture and in animal models [99,100]. It has been extended to a phase I/II trial for patients with a variety of EBV-positive malignancies [101] with some success. One conclusion from these studies is that events downstream of EBV’s DNA synthesis during its lytic phase do not contribute detectably to its oncogenesis once cells have been infected and transformed. A second is that its lytic DNA synthesis does not contribute to the cancer phenotypes of these cells either.

We lack small-molecule inhibitors of steps earlier than DNA synthesis for EBV’s lytic phase. Another window, though, on the potential contributions of these steps to EBV’s oncogenesis comes from detailed studies of treating tumor patients with T cells educated against EBV-encoded antigens. Researchers have developed adoptive T-cell therapies to treat EBV-positive malignancies successfully [102,103]. In general, the epitopes that have been recognized by the cytotoxic T cells are expressed during the latent phase of EBV’s life cycle, in part reflecting the viral gene expression of LCLs used to educate the T cells in vitro. Some of these studies have also been conducted with T cells educated with a pool of peptides including some derived from genes expressed during the early portion of EBV’s lytic phase [104]. These T cells educated to recognize epitopes encoded by BMLF1, BRLF1, and BZLF1 which are expressed early in EBV’s lytic phase are functional in vitro (ibid.). They have not yet been shown alone of being capable of limiting EBV-positive malignancies so that an essential role for these genes in EBV’s oncogenesis remains unclear. These epitope-specific, anti-EBV T cells are, however, tools that should allow testing for contributions of the viral genes expressed early in EBV’s lytic phase to EBV’s oncogenesis, particularly in tractable animal models. Such tools, therefore, can be instrumental in learning how these early lytic genes foster EBV’s oncogenesis.

## 8. Concluding Remarks

EBV’s lytic phase contributes to tumorigenesis primarily in two ways (see Figure 1): (1) the production of infectious particles to infect more cells, and (2) the regulation of cellular oncogenic pathways, mediated by lytic proteins and miRNAs. The production of infectious virus is a requisite precursor to the infection and transformation of cells that can subsequently evolve into tumors. Following infection, reactivation of the lytic phase supports the expression of miRNAs and early lytic genes which can regulate cellular pathways that promote tumorigenesis. Some of these tumorigenic effects are cell autonomous, affecting only the cells in which the relevant lytic genes are expressed. Others are non-cell autonomous, exerting influence over neighboring tumor cells through the production of secreted molecules and/or the modification of the tumor microenvironment. As the completion of a lytic phase results in cell death, we speculate that the contribution of lytic phase to tumorigenesis is in part mediated by an incomplete lytic phase (also termed abortive lytic phase), highlighting the importance of the early lytic phase. Given the relevance of the lytic phase to tumor progression and maintenance, it will be important to understand the mechanisms by which EBV’s lytic phase contributes to tumorigenesis in order to target it as an alternative means to treat EBV-associated malignancies.

## Figures and Tables

**Figure 1 microorganisms-08-01824-f001:**
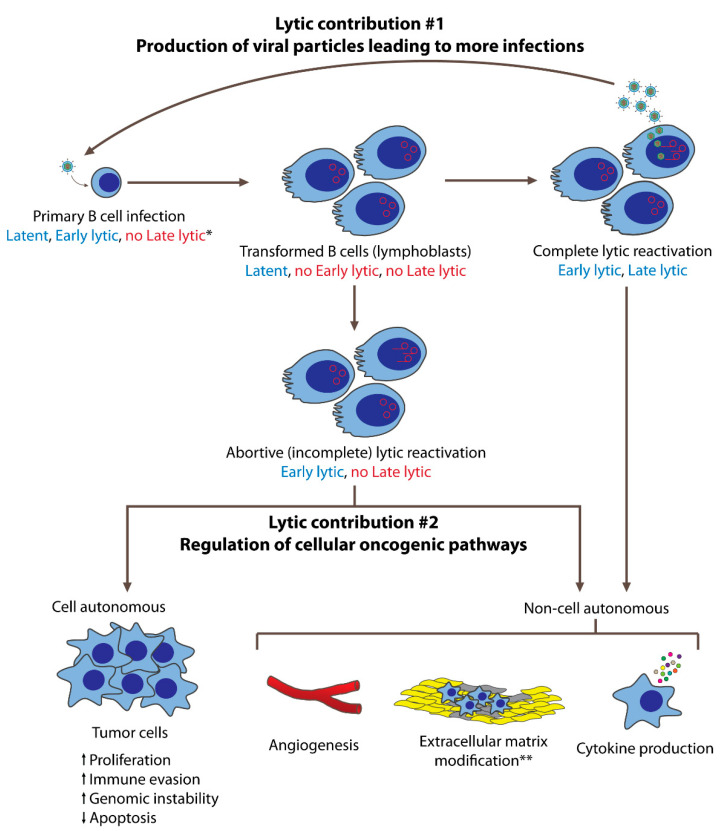
EBV’s lytic phase contributes to oncogenesis in both B cells and epithelial cells. In this figure, B cell lymphomagenesis is used to represent these contributions. Following primary infection and transformation, EBV is maintained latently in infected cells. On rare occasions, some cells undergo lytic reactivation, either completely or incompletely. Cells that undergo the complete lytic reactivation express both early and late lytic genes and produce new viral particles that can infect more cells. These newly infected cells are transformed and may subsequently evolve into tumor cells. Cells that complete the lytic phase eventually die, such that they do not contribute as proliferating tumor cells. However, they can contribute to oncogenesis via non-cell-autonomous mechanisms mediated by early lytic gene products. These contributions include angiogenesis, pro-tumorigenic cytokine production, and, in the case of NPCs, extracellular matrix modifications. Some cells that enter the lytic phase do not complete it, undergoing incomplete (abortive) lytic reactivation. These cells express early lytic genes but not late lytic genes, and thus do not produce new viral particles. These cells may continue to live, and contribute to oncogenesis cell autonomously by becoming tumor cells with increased proliferation, immune evasion, and genomic instability, as well as decreased apoptosis. Having expressed early lytic genes, abortive lytic cells may also contribute to oncogenesis non-cell autonomously. * Expression of latent, early lytic, or late lytic genes are indicated in blue (expressed) and red (not expressed). ** Extracellular matrix modifications are primarily studied in NPCs.

**Table 1 microorganisms-08-01824-t001:** EBV’s lytic genes and their roles in oncogenesis.

Immunomodulation and Immune Evasion					
EBV Lytic Gene	IE/E/L ^1^	Lytic Function	Role in Oncogenesis	Oncogenic Mechanism of Action	References
BZLF1	IE	Transactivator	Induction of pro-inflammatory cytokine expression and secretion (IL-8, IL-10, IL-13)	Binding and activating target gene promoters	[49,50,51]
BGLF5	E	Alkaline exonuclease	Downregulation of MHCs	Host shut off; degradation of cellular mRNAs	[52]
BILF1	E	gp64, vGPCR	Inhibition of MHC trafficking		[53]
BLLF3	E	dUTPase	Induction of pro-inflammatory cytokine expression and secretion (IL-1β, IL-6, IL-8, IL-10)		[54]
BNLF2a	E	Inhibitor of TAP ^2^	Inhibition of CD8 T cell recognition of infected cells		[55]
BCRF1	L	vIL-10	Inhibition of NK cell-mediated elimination of infected cells; inhibition of CD4 T cells		[55]
BDLF3	L	gp150	Downregulation of MHCs	Ubiquitination and degradation of MHCs	[56]
BZLF2	L	gp42	Inhibition of MHC II-mediated antigen presentation		[57]
**Angiogenesis and Invasion**					
**EBV Lytic Gene**	**IE/E/L**	**Lytic Function**	**Role in Oncogenesis**	**Oncogenic Mechanism of Action**	**References**
BZLF1	IE	Transactivator	Upregulation of MMP1, MMP3, MMP9	Binding and activating target gene promoters	[58,59,60]
BRLF1	IE	Transactivator	Upregulation of MMP9	Binding and activating target gene promoters	[61]
***Genomic Instability***					
**EBV Lytic Gene**	**IE/E/L**	**Lytic Function**	**Role in Oncogenesis**	**Oncogenic Mechanism of Action**	**References**
BALF3	E	Terminase	Induction of genomic aberration	Induction of DNA damage	[62]
BGLF4	E	S/T protein kinase	Induction of genomic aberration	Induction of DNA damage pathways and premature chromosome condensation	[63,64]
BGLF5	E	Alkaline exonuclease	Induction of genomic aberration	Induction of DNA damage	[65]
BALF4	L	gp110	Induction of genomic aberration		[66]
BNRF1	L	Major tegument protein	Induction of genomic aberration		[66]
**Cell Cycle Progression and Apoptosis**					
**EBV Lytic Gene**	**IE/E/L**	**Lytic Function**	**Role in Oncogenesis**	**Oncogenic Mechanism of Action**	**References**
BALF1	E	vBcl-2	Pro-survival, anti-apoptotic		[67]
BHRF1	E	vBcl-2	Pro-survival, anti-apoptotic	Inhibition of BIM, PUMA, BAK	[67,68,69]

^1^ Lytic gene classifications; IE: immediate early; E: early; L: late. ^2^ TAP: transporter-associated with antigen processing.

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
