# Peer review of "Epstein–Barr Virus: How Its Lytic Phase Contributes to Oncogenesis"

_microorganisms, 2020, doi:10.3390/microorganisms8111824_

Round 1
Reviewer 1 Report
The authors clearly show how lytic phase of EBV contribute to EBV oncogenesis. I also agree the importance of lytic infection of EBV in oncogenesis.
I have only one request to the authors.
Almost 20 years ago, some researchers such as Dr. Irene Joab’sgroup showed that EBV also associates with oncogenesis of breast cancers. I think some ms shows ZEBRA expression is important in EBV-related breast cancers. It is still contravertial whether EBV is related with pathogenesis of breast cancers. However, I feel sure EBV related with pathogenesis of breast cancers at some extent, after I read this review. If the authors permit these hypothesis, I am glad if they put some reference and description about EBV-related breast cancers and ZEBRA expression.
Author Response
We thank reviewer #1 for their thoughtful comments and suggestions.
Reviewer #1 Comments
The authors clearly show how lytic phase of EBV contribute to EBV oncogenesis. I also agree the importance of lytic infection of EBV in oncogenesis.
I have only one request to the authors.
Almost 20 years ago, some researchers such as Dr. Irene Joab’sgroup showed that EBV also associates with oncogenesis of breast cancers. I think some ms shows ZEBRA expression is important in EBV-related breast cancers. It is still contravertial whether EBV is related with pathogenesis of breast cancers. However, I feel sure EBV related with pathogenesis of breast cancers at some extent, after I read this review. If the authors permit these hypothesis, I am glad if they put some reference and description about EBV-related breast cancers and ZEBRA expression.
Response to Reviewer #1
We agree that the association of EBV with breast cancer would be an interesting topic of discussion. Our group, in collaboration with our colleagues, have investigated the presence and potential contribution of EBV in breast cancer (Perrigoue et al. 2005 Cancer Epidemiol Biomarkers Prev). We assayed 45 matched tumor and normal breast biopsy samples using RT-qPCR with probes against two EBV DNA targets, and found that none of the tumor and normal tissue sample had detectable EBV genomes. IHC for EBERs on a subset of matched samples confirmed the absence of EBV genome in breast carcinomas. Given the controversial nature of the presence of EBV in breast cancer, we regarded this topic to require an extensive discussion and thus to be outside the scope of this review on the role of lytic phase in EBV oncogenesis. We do thank the reviewer for the thoughtful suggestion.
Reference:
Perrigoue JG, den Boon JA, Friedl A, Newton MA, Ahlquist P, Sugden B. Lack of association between EBV and breast carcinoma. Cancer Epidemiol Biomarkers Prev. 2005 Apr;14(4):809-14. doi: 10.1158/1055-9965.EPI-04-0763. PMID: 15824148.
Reviewer 2 Report
The contention of an oncogenic role for EBV lytic genes has for years faced resistance from the fundamental cell lysis feature of the lytic cycle, garnering the dogma of latency genes as the primary driver of oncogenesis. Nevertheless, this resistance has been whittled away through gradual yet persistent investigations from a number of groups including many key studies by the Kenney lab in mouse models. This review catalogues various and sundry studies over the years in a well informed, thoughtful, and logical treatment of this topic of growing importance to our understanding of lytic factors in EBV mediated oncogenesis. The review aptly addresses both direct involvements of lytic factors as well as the possible role of increased load of infected cells through produced virus particles. Also included in this review is the interesting topic of a possible role of increased genomic instability induced by various viral lytic factors and interestingly, viral like particles. The review is nicely concluded with clinically oriented findings utilizing chemical inhibitors. Together, this is a well crafted review on the topic of lytic factors in EBV mediated oncogenesis. A few minor specific points:
- The sentence beginning on line 84, “These combined findings indicate…and this increase arose from the release of EBV from infected cells leading to new infection of uninfected B cells.” Should this be tempered to account for the possibility that the increases in the infected pool could have occurred through latency associated expansion under immunocompromised settings? “…and this increase may have arisen from…”?
- A reference should be included for studies alluding to the expression of late genes in tumor samples (line 154).
- In reference to the late genes detected in tumor samples, it is indicated that this is a conundrum since viral replication factor expression was not detected. The explanation that this might be due to low sensitivity for detecting these. Nevertheless, is it possible that some of these are in the class of “leaky late” genes as reported by Eric Johannsen? In this case, the late genes might be expressed, perhaps at somewhat low levels, in the absence of viral DNA replication.
- Line 222. Perhaps the sentence, “An increased level of…” could be altered to something like, “In wild type infected cells, an increased level of…”. As is, the topic is the dBALF5 virus and saying that the increased level could be due to viral replication adds temporary confusion.
- In the paragraph of lines 217-234, the authors are providing evidence that blocking DNA replication may play a role in increasing oncogenesis. Do the authors want to include possible clinical support for this from studies showing farily frequent deletion of the right side of the BamHI A locus in T/NK-cell lymphomas ("Defective Epstein-Barr virus in chronic active infection and haematological malignancy". Nat Microbiol. 2019 Mar;4(3):404-413. doi: 10.1038/s41564-018-0334-0. Epub 2019 Jan 21.) that might include the deletion of viral DNA replication factors? There may also be other studies along these lines as well.
- Line 120. “Overtime…” should be “Over time…”
Author Response
We thank the reviewer for their gracious comments. We appreciate the reviewer's thoughtful suggestions and have modified the text accordingly. Here we list our responses to each of the reviewer's comments.
Reviewer #2 Comments
The contention of an oncogenic role for EBV lytic genes has for years faced resistance from the fundamental cell lysis feature of the lytic cycle, garnering the dogma of latency genes as the primary driver of oncogenesis. Nevertheless, this resistance has been whittled away through gradual yet persistent investigations from a number of groups including many key studies by the Kenney lab in mouse models. This review catalogues various and sundry studies over the years in a well informed, thoughtful, and logical treatment of this topic of growing importance to our understanding of lytic factors in EBV mediated oncogenesis. The review aptly addresses both direct involvements of lytic factors as well as the possible role of increased load of infected cells through produced virus particles. Also included in this review is the interesting topic of a possible role of increased genomic instability induced by various viral lytic factors and interestingly, viral like particles. The review is nicely concluded with clinically oriented findings utilizing chemical inhibitors. Together, this is a well crafted review on the topic of lytic factors in EBV mediated oncogenesis. A few minor specific points:
- The sentence beginning on line 84, “These combined findings indicate…and this increase arose from the release of EBV from infected cells leading to new infection of uninfected B cells.” Should this be tempered to account for the possibility that the increases in the infected pool could have occurred through latency associated expansion under immunocompromised settings? “…and this increase may have arisen from…”?
Response:
We modified the text to reflect the contribution of expansion of latently infected cells to the increases in the infected pool.
- A reference should be included for studies alluding to the expression of late genes in tumor samples (line 154).
Response:
We added references of studies in which the expression of late genes was detected in tumor samples despite evidence for an incomplete lytic phase:
- Borozan, I.; Zapatka, M.; Frappier, L.; Ferretti, V. Analysis of Epstein-Barr Virus Genomes and Expression Profiles in Gastric Adenocarcinoma. J. Virol. 2017, 92, 1–18, doi:10.1128/jvi.01239-17.
- Martel-Renoir, D.; Grunewald, V.; Touitou, R.; Schwaab, G.; Joab, I. Qualitative analysis of the expression of Epstein-Barr virus lytic genes in nasopharyngeal carcinoma biopsies. J. Gen. Virol. 1995, 76, 1401–1408, doi:10.1099/0022-1317-76-6-1401.
- In reference to the late genes detected in tumor samples, it is indicated that this is a conundrum since viral replication factor expression was not detected. The explanation that this might be due to low sensitivity for detecting these. Nevertheless, is it possible that some of these are in the class of “leaky late” genes as reported by Eric Johannsen? In this case, the late genes might be expressed, perhaps at somewhat low levels, in the absence of viral DNA replication.
Response:
We have modified the text to reflect the possibility that the leaky late genes underlie the apparent expression of late genes in samples lacking lytic DNA replication. Indeed, one of the detected late genes, BALF4, was identified by Eric Johannsen’s group as a leaky late gene. Thus, it is possible that BALF4 was detected in tumor samples due to its low-level expression in the absence of lytic DNA replication. However, other late genes (BLLF1, BNRF1, BPLF1) detected to be expressed in the tumor samples have been identified as true late genes, meaning their expression is dependent upon lytic DNA replication. In this latter case, the conundrum remains, the answer to which might be that the required DNA replication genes were detected inefficiently and/or were only expressed in a small subset of the tumor cells.
- Line 222. Perhaps the sentence, “An increased level of…” could be altered to something like, “In wild type infected cells, an increased level of…”. As is, the topic is the dBALF5 virus and saying that the increased level could be due to viral replication adds temporary confusion.
Response:
We have modified the text as suggested to clarify that it refers to cells infected with wildtype EBV.
- In the paragraph of lines 217-234, the authors are providing evidence that blocking DNA replication may play a role in increasing oncogenesis. Do the authors want to include possible clinical support for this from studies showing farily frequent deletion of the right side of the BamHI A locus in T/NK-cell lymphomas ("Defective Epstein-Barr virus in chronic active infection and haematological malignancy". Nat Microbiol. 2019 Mar;4(3):404-413. doi: 10.1038/s41564-018-0334-0. Epub 2019 Jan 21.) that might include the deletion of viral DNA replication factors? There may also be other studies along these lines as well.
Response:
We have modified the text to include the suggested finding regarding intragenic deletions of regions essential for DNA replication and infectious viral particle production in T/NK-cell lymphomas.
- Line 120. “Overtime…” should be “Over time…”
Response:
We modified the text accordingly.